## Evaluation of intervention fidelity of a complex psychosocial intervention Lifestyle Matters: a randomised controlled trial

Kirsty Sprange ,[1] Gail Mountain,[2] Claire Craig[3]

► Prepublication history and additional supplemental materials for this paper is available online. To view these files, please visit the journal online (http://dx.doi.org/10. 1136/bmjopen-2020-043478).

¹Nottingham Clinical Trials Research Unit, University of Nottingham, University Park, Nottingham, UK
²Centre for Applied Dementia Studies, Faculty of Health Studies, University of Bradford, Bradford, UK
³Lab4Living, Sheffield Hallam University, Sheffield, UK

**Correspondence to**
Kirsty Sprange;
Kirsty.Sprange@nottingham.ac.uk

## ABSTRACT

**Objectives** Robust research of complex interventions designed to promote mental well-being in later life is required to inform service development. An essential component is ensuring that such interventions are delivered as intended. We present a detailed description of the design and implementation of a fidelity assessment within a trial of one such intervention (Lifestyle Matters). The findings help to explain the trial results and also inform the design of embedded fidelity assessments within future evaluations of complex interventions.

**Design** We conducted a mixed-method fidelity assessment embedded as part of a multicentre pragmatic randomised controlled trial. A conceptual fidelity framework was developed from the Behaviour Change Consortium framework. From this the fidelity assessment was designed. The resulting instrument assessed the following parameters: intervention design, training, supervision; and delivery, receipt and enactment of the intervention.

**Intervention** The Lifestyle Matters intervention was designed to assist older people to improve and sustain mental well-being through participation in meaningful activity. The aim is to enable participants to engage in both new and neglected activities through a mix of facilitated group meetings and individual sessions.

**Results** The fidelity assessment demonstrated that the intervention was delivered as per protocol for the group component and was tailored to meet individual needs. There was substantial inter-rater agreement for training; and group member performance 0.72; and moderate agreement for facilitator performance 0.55. It was not possible to determine whether small declines seen in facilitator performance were due to facilitator drift or moderating factors such as group dynamics or participant characteristics.

**Conclusions** The assessment methods adequately measured criteria identified as being significant indicators of fidelity. Adherence during training, delivery and supervision was good. The subjective nature of identification and rating observed behaviours was the main challenge. Future research should explore alternative methods of assessing fidelity in trials of complex interventions.

**Trial registration number** ISRCTN67209155.

### Strengths and limitations of this study

► This study demonstrated that it is feasible to design a fidelity assessment framework and develop bespoke assessment tools to evaluate a complex multicomponent psychosocial intervention.
► The extent of the assessment was limited by the methods used, for example it was not appropriate or even ethical to undertake observations of activities in the community.
► The validity of the findings is reliant on whether bespoke assessment tools are sensitive enough to identify the complex constructs within this type of intervention.

## BACKGROUND

An increase in long-term conditions within the global population combined with increased longevity has led to interest in how tailored psychosocial interventions might enable individuals to retain physical and mental well-being in the extended life span.[1 2] UK guidance has identified the need for multicomponent, tailored individual and group-based preventative interventions which promote health and well-being through occupation and physical activity.[3] The benefits of social networking and group interventions involving activity targeting social isolation and loneliness has also been identified[4] but there is a limited range of recommended interventions. The challenges of associated research include intervention development and evaluation of intervention acceptability, followed by robust testing to determine intervention effectiveness.[3] Robust testing should include an assessment of intervention fidelity to assess the extent to which the intervention is delivered as intended.

Fidelity assessment and maintenance of fidelity[5] enables researchers to demonstrate with greater confidence that study results were due to the intervention and not

to other confounding factors. It is acknowledged that using multiple methods as part of a fidelity assessment can enhance validity and reliability of complex interventions.[6] Development and application of such multimethod fidelity assessments in complex multicomponent interventions have been demonstrated as feasible.[7–9]

A pragmatic randomised controlled trial (RCT) with an embedded fidelity assessment was conducted to evaluate a complex manualised intervention, Lifestyle Matters, which was created to promote the mental well-being of community living older people.[10] Lifestyle Matters was developed for the UK setting from a US intervention, Lifestyle Redesign which was found to promote physical and mental health and well-being, occupational functioning and life satisfaction in older adults.[11 12] As reported in Mountain *et al*[10] the trial fully adhered to Consort guidelines. There is some debate about the benefits or otherwise of the manualisation of complex interventions but one argument suggests that manualisation can reduce variance in intervention delivery thereby enhancing fidelity.[13]

Fidelity assessment for the Lifestyle Matters study included an evaluation of facilitator training and intervention delivery against predetermined criteria; and a complementary qualitative exploration of the views of facilitators regarding how lifestyle change might be supported or not through the intervention and their views of how they had been able to deliver this goal.[14]

Medical Research Council (MRC) guidance recommends that complex intervention design and associated fidelity assessment frameworks are developed from the underlying intervention theory.[15] Lifestyle Matters is underpinned by social cognitive theory. There is a focus on promotion of self-management by participants,[16] and in particular on developing and sustaining self-efficacy as this is known to be a protective factor in improving motivation and emotional well-being.[17] Intervention theory is located in the promotion of self-efficacy[18] as well as a model that proposes that the transactional relationships between the person, environment and activities influence occupational performance. Occupation in this context refers to meaningful activity and active participation in life.[19] The intervention guides participants to identify their own personal goals and work towards them. They are encouraged to use peer support through the group and support from facilitators in both group and individual sessions to actively overcome barriers to meaningful participation within their communities.

Interventions like Lifestyle Matters will have core components such as number of group and individual sessions and structured manual topics. However adapting a complex intervention to meet the needs of individual participants is important.[15 20] The Lifestyle Matters intervention provides a content and delivery structure, which is both flexible and readily tailored to the needs of participants. Therefore, a range of methods and tools to assess and evaluate the fidelity of this tailored intervention was required. This was important as the application of measures to assess intervention adaptation can improve our understanding of how different components can impact on delivery and receipt in context.[5 21]

This paper describes the fidelity assessment conducted for the Lifestyle Matters study and presents the findings from analysis of facilitator training and supervision, intervention delivery and receipt. The findings are then discussed in the context of intervention implementation.

## Trial design

The study was a pragmatic, two-arm, parallel group, individually RCT designed to determine the population benefit of an occupation-based intervention for people aged 65 years or older.[10] A total of 288 participants was randomised between 14 August 2012 and 19 April 2013 (18 couples and 252 individuals); 145 and 143 were allocated to the intervention and control groups, respectively. The intervention was delivered in one city in the north of England and in North Wales. Participants were eligible if they had reasonable cognitive function to participate in the intervention as evidenced by a score of 0 to 7 on the Six Item Cognitive Impairment Test.[22] They were also living independently or in sheltered accommodation, alone or with others and were able to converse in English or Welsh. An associated fidelity assessment and qualitative substudy were conducted to evaluate facilitator training and supervision; intervention delivery; and intended changes to lifestyle by participants. The latter are reported elsewhere.[14 23]

## The Lifestyle Matters intervention

The intervention involves volunteers taking part in 16 facilitated sequential weekly group meetings with up to 15 others and in four individual sessions with one of the facilitators to pursue individual goals.[10 24] During group meetings participants are encouraged to think about and engage in discussion and activities to explore the relationship between occupation and health, selecting from a menu of potential topics. A crucial element is going into the community to try out new or neglected skills with support from each other and from the facilitators; at least four sessions should involve such activities.

Group meetings are approximately 2 hours in duration at a convenient community venue. The first group sessions are structured to cover key topics from the manualised intervention. This provides a foundation for facilitators to maintain adherence to the intervention and for both facilitators and participants to work together on future sessions using the same model. Short breaks of no more than 2 weeks can be included to accommodate statutory holidays.

Individual sessions are an important intervention component and are specifically directed by the individual's goals and preferences. The first session concentrates on the participant-facilitator relationship. The remaining three sessions are built around the participant and what they want to achieve, using the intervention to assist them to execute this.

## Facilitators and facilitation

Facilitators with the necessary skill set and qualities were recruited to deliver the intervention as intended.[25] A job description was produced which detailed core requirements, deemed to be essential for understanding the intervention and enabling successful delivery. These included prior experience of working with groups and with older people in community settings and specifically the planning and delivery of group and individual sessions. A requirement of intervention delivery is for two facilitators to be present during each group and to share working with participants during individual sessions. Facilitators could be involved in the delivery of up to three intervention groups of 8–16 participants at any one time (including associated individual sessions).

## Facilitator supervision

Supervision was provided by appropriately experienced Occupational Therapists who also attended facilitator training alongside the facilitators. A protocol was created to enable provision of consistent and appropriate supervision across and within sites. Regular one-to-one supervision was recommended on a weekly basis at a mutually convenient time and place, preferably face-to-face but with distance supervision being an option if appropriate. Joint supervision was also deemed acceptable if the individual supervisory needs of facilitators had been met.

## Training

A 2-day intensive training course was delivered jointly to all facilitators and supervisors before intervention delivery commenced, thereby ensuring that everyone received the same information and training experience to equip them to deliver the intervention and improve fidelity.[26]

Training content included presenting the rationale and theory behind the intervention and intervention components.[25] The training then mirrored delivery of the Lifestyle Matters intervention and manual,[27] in that it involved;

► Understanding the importance of taking a personalised and tailored approach towards participants at all times.[28]
► Didactic sessions to explain the programme and its ethos.
► Peer sharing; providing the opportunity to share narratives, explore experience and attitudes, and reflect on the relationship that can develop between participants and between facilitators and participants and between supervisor and facilitators.
► Undertaking active experimentation, thereby testing out ideas and concepts.
► Exploring communication skills through role playing, vignettes and group discussions.

The 2-day training was video recorded to give facilitators an idea of how they and their participants might experience this for fidelity assessment and smooth out any challenges with using the recording equipment.

All facilitators and supervisors received a copy of the manual and an accompanying CD-ROM with downloadable resources for intervention delivery.[27]

## MATERIALS AND METHODS

### Fidelity assessment framework

A comprehensive fidelity assessment framework specific to the intervention was developed prior to intervention delivery (see table 1) based on the Behaviour Change Consortium (BCC) recommendations and National Institute of forHealth and Care Excellence guidance on behaviour change.[29 30] Fidelity assessment and quality assurance parameters centred on the five BCC recommended criteria: intervention design, training, delivery, receipt and enactment. For each criterion, programme objectives based on the Lifestyle Matters manualised intervention and training were discussed and agreed by two senior researchers. A description for each standard was then developed and methods to demonstrate fidelity to each standard identified (see table 1).

### Assessment tools

The underlying ethos of Lifestyle Matters and the tailored approach to delivery make fidelity assessment complex as many of the identified criteria are subjective. A range of quantitative and qualitative tools was therefore used in line with MRC recommendations for complex interventions,[15] including attendance registers, observation checklists and interview schedules which were all piloted before application.

### Intervention attendance registers

Registers were used to routinely record both group and individual session attendance. Using pilot study findings it was estimated that participants would need to attend at least half (eight) group meetings, to benefit from the intervention.[31] The number of individual sessions required to derive benefit was not set.

### Development of observation checklists

Observation checklists were developed by two senior researchers who identified domains comprising the core skills and key criteria identified in the Lifestyle Matters manual, worksheets and training materials (see online supplemental additional file 1). Criteria described the components and behaviours expected to be present during intervention delivery. Domains were divided into those associated with 'Trainer or Trainee performance' and those associated with 'Facilitator or Group member performance'.

A scoring system was devised. Several criteria were rated on a binary 'Yes' or 'No' scale for presence or absence. For the purpose of analysis if a criterion was present this was converted to a score of 3. If the criterion was not present this was converted to a score of 0.

**Table 1** Lifestyle Matters fidelity assessment framework and results

| Element | Fidelity criteria | Fidelity standard | Standard met | Evidence |
|---|---|---|---|---|
| **Trial design** | | | | |
| Comparable treatment | All participants received the same programme tailored to the needs of the group/setting. | ▲ Sixteen weekly meetings offered to all participants<br>▲ Attendance recorded | Yes | ▲ Trial documentation, for example, information sheet<br>▲ 176 (11×16) weekly registers recorded |
| | | ▲ Four individual sessions offered to all participants<br>▲ Attendance recorded<br>▲ Four out-of-venue sessions offered to all participants<br>▲ Attendance recorded | Yes | ▲ Trial documentation, for example, information sheet<br>▲ 443 individual sessions offered, 143 taken up (32%)<br>▲ On average participants took part in four out-of-venue activities across the 16 weeks |
| Risk to implementation | Plan for potential issues that could affect the delivery of the Lifestyle Matters programme | ▲ A range of recruitment strategies will be implemented | Yes | ▲ Methods used included GP mail-outs, direct referrers and posters/leaflets |
| | | ▲ A range of days and times for meetings offered, from which participants can choose | Yes | ▲ Facilitators worked flexibly<br>▲ Groups took place on various days of the week and times of day (AM/PM) |
| | | ▲ Undertake three recruitment cycles in three geographical areas, to prevent saturation | Yes | ▲ Recruitment conducted in multiple areas at each site |
| **Monitoring provider training** | | | | |
| Standardised training and facilitator skill acquisition | All facilitators receive the same training tailored to the group/setting and engage with the training in a similar way | ▲ Completion of training exercise by facilitators | Yes | ▲ Observation by researchers of participants completing role playing exercises and group work |
| | | ▲ Training delivered by the same trainer | Yes | ▲ Same trainer delivered all training |
| | | ▲ Manual/CD-ROM provided to all trainees | Yes | ▲ Manual (or copy) and CD provided to all trainees |
| | | ▲ Observation of the training session by two researchers using a content checklist | Yes | ▲ Observation by two researchers of skill transference using content checklist<br>▲ Checklist coding developed to observe 'All' participants |
| **Monitoring intervention delivery** | | | | |

Continued

**Table 1** Continued

| Element | Fidelity criteria | Fidelity standard | Standard met | Evidence |
|---|---|---|---|---|
| Standardised delivery | All facilitators using the same techniques and content from the programme | ▶ Observation using a content checklist by two researchers | Yes | ▶ Observation completed by two researchers |
| | | ▶ 75% of opportunities for completing goal setting are recorded (for individual and group) | No | ▶ Participants found concept of 'goal setting' difficult to understand. 'Goal setting' sheets provided in programme rarely used. Discussion of 'Planning' and 'Wishes' preferred |
| | | ▶ Range of materials from the Lifestyle Matters programme received by all participants | Yes | ▶ Use of materials discussed by participants during qualitative interviews<br>▶ Observation of manual materials used during recorded group meetings |
| | | ▶ Facilitators maintain reflective diaries | Yes | ▶ Used as part of supervision sessions, confirmed by facilitators and supervisors |
| | | ▶ Registers completed for weekly meetings and individual sessions | Yes | ▶ Registers completed and recorded on the trial database |
| | | ▶ Participant and facilitator semistructured interview topic guide | Yes | ▶ Interviews conducted with all facilitators at the end of cycle 1 and cycle 3 Interviews conducted with 10% sample of participants at the end of their programme attendance |
| | | ▶ All participants receive certificate of attendance/achievement | Yes | ▶ Copies of certificates given to all participants across both sites |
| | | ▶ Facilitators meet the National Health Service Band 4 equivalent job description | Yes | ▶ Facilitators selected for interview based on meeting essential criteria of job description. |
| Minimise drift in skills/delivery | Adherence to training content and delivery over the three cycles of the intervention | ▶ Observation using a content checklist by two researchers | Yes | ▶ Observation by two observers |
| | | ▶ OT supervisor protocol | Yes | ▶ Protocol written and provided to supervisors and facilitators prior to the programme start |
| | | ▶ Each facilitator will attend between 8 and 16 supervision sessions in total of which half should be delivered face to face | Yes | ▶ Interviews: Reported face-to-face contact recorded for one facilitator for 12 months. Three facilitators' supervision started as face to face moving to a combination with telephone contact<br>▶ Supervisor attendance records provided for two facilitators (consent obtained) |
| Monitoring receipt of intervention | | | | |

Continued

**Table 1** Continued

| Element | Fidelity criteria | Fidelity standard | Standard met | Evidence |
|---|---|---|---|---|
| Participant attendance and engagement | Recording participants' weekly attendance<br>All participants taking part in the meetings and activities<br>Impact of intervention on participant in terms of well-being | ▲ Registers completed by facilitator for weekly meetings and individual sessions where arranged | Yes | ▲ Interviews demonstrated that most participants took part in the programme, although not all joined every activity mainly due to health or mobility issues<br>▲ All registers completed and recorded |
| | | ▲ 75% of opportunities for completing goal setting are recorded (individual and group) | No | ▲ Concept of 'goal setting' difficult to understand by participants. 'Planning' or 'Wishes' preferred |
| | | ▲ Participant and facilitator semistructured interview topic guide | Yes | ▲ 10% sample of participants interviewed<br>▲ Participant behaviours observed during videoed sessions<br>▲ All facilitators interviewed at cycle 1 and cycle 3 |
| | | ▲ PROMS | Awaiting results | ▲ Completed PROMS recorded on trial database |

Adapted from Bellg et al.[29]
GP, General Practitioner; OT, Occupational Therapist; PROMS, Patient Reported Outcome Measures.

Fidelity scores were calculated based on agreement of a final score between the two coders and the total was given a percentage score as follows. The scoring system was adapted from the principle set out in Borrelli, that 80%–100% constitutes high fidelity:[25]

▶ 60% Unsatisfactory.
▶ 61%–70% Satisfactory.
▶ 71%–80% Good.
▶ 81%–90% Very good.
▶ 91%–100% Excellent.

To ensure consistency the same two researchers completed fidelity assessment activities and coding. The patient and public contributor for the project also contributed to the fidelity assessment analysis including reviewing and piloting observation checklists (as described below), as well as participant observation during training.

### Assessment of the fidelity of training delivery

Participant observation, in which the researcher not only observes, but takes an active role in the setting,[32] was selected to assess the fidelity of training delivery. The researchers therefore attended the training as delegates. This decision was taken as participant observation allows the observer to not only have the same experience as those taking part, as an insider, but also enables them to understand the more subtle aspects of delivery such as group dynamics as an outsider.[33] However, this can also result in loss of perspective and objectivity to some extent.

During the training, sessions were independently rated by the two researchers. There were also discussions between the trainer and researchers at the beginning and end of each day to review content and feedback from attendees. All scores were then shared and reviewed for reliability of interpretation and coding against checklist criteria.

### Assessment of the fidelity of intervention delivery

A purposive sample of eight video recordings was taken of 4 of the 11 Lifestyle Matters groups to monitor implementation and adherence to the intervention and identify any facilitator drift over time. To reduce bias the same number of groups and group meetings from both sites were selected for assessment. Written consent to conduct video recordings was obtained from each participant and facilitators recorded the group meetings using simple compact video recorders.

One video of each of the four selected groups was recorded at approximately week 4 and again at around week 10 of intervention delivery although some flexibility in session choice was necessary to avoid out-of-venue activities (see table 2). Observing two meetings per group, one at the beginning and one near the end, enabled us to examine any learning effects and identify any facilitator drift.[34]

Video content was then rated using the predetermined checklist and findings analysed to determine delivery and receipt of the intervention and any change over time.

**Table 2** Sample of video-recorded group meetings

| Site | Wave | Lifestyle Matters group | Programme week (1–16) videoed | No of facilitators present | Participants attending/registered |
|---|---|---|---|---|---|
| North England | 1 | 1 | 6 | 2 | 10/13 (77%) |
| | | | 10 | 2 | 11/13 (85%) |
| North Wales | 1 | 1 | 4 | 2 | 7/10 (70%) |
| | | | 10 | 2 | 7/9* (78%) |
| North England | 3 | 6† | 7 | 2 | 6/8 (75%) |
| | | | 15 | 2 | 7/8 (88%) |
| North Wales | 3 | 5† | 4 | 2 | 8/9 (89%) |
| | | | 10 | 2 | 7/9 (78%) |

*Participant withdrew.
†North England site completed six Lifestyle Matters groups and North Wales completed five.

During piloting of the intervention fidelity checklist, it became apparent that several of the assessment criteria could not be observed during the weekly group meetings. For example, receipt and enactment of the intervention would occur in the participant's everyday life away from the group. It was deemed that either the criterion could not be present during every weekly meeting or it would be necessary to observe several meetings consecutively which was not planned. Further observations using the checklist to test the assumption found this to be consistent. The criteria deemed to be immeasurable were therefore excluded from the final analysis.

Due to practicalities, only the groups held within the meeting venue were observed. Given the importance of enactment of activities in the community this was a limitation.

We did not undertake observations of the individual sessions as these were deemed a safe space for participants to work with their facilitator on individual goals. These sessions could also involve activities in the community where it would be impractical and unethical for researchers to observe sessions.

### Inter-rater reliability

Inter-rater reliability demonstrates whether available data accurately represent the variables being measured and the extent to which coders attribute the same score to the same variable.[35] Inter-rater reliability was calculated by two researchers independently coding video recordings and by scores being compared using the Kappa statistic, with a value of 0.61 and above indicating good inter-rater agreement.

► <0.20 Poor.
► 0.21–0.40 Fair.
► 0.41–0.60 Moderate.
► 0.61–0.80 Good.
► 0.81–1.00 Very good.

### Semistructured interviews

Semistructured interviews were undertaken with all facilitators to explore views on what enables effective intervention facilitation, and the skills and competencies required to deliver the intervention and enable participants to act on it.[14 23] Interviews took place at the end of the first and third (final) waves of intervention delivery to identify any changes in experiences of delivery over time.

An interview topic guide was developed informed by the content of the training materials and intervention manual as well as knowledge gained from prior feasibility work.[31]

A purposeful sample of 10% of participants was interviewed, selected from both sites and from each of the three waves of intervention delivery to elicit the range and nature of issues that influenced their experiences of the intervention. The sample was considered adequate to reach data saturation within resource limitations.[36]

All transcripts were read for familiarisation and annotated for themes by the same two researchers (fidelity lead and second researcher) using framework analysis in NVivo.[37 38] For the purposes of the fidelity assessment, data from the interviews were used to identify cross-cutting themes and explanatory factors for fidelity outcomes.

### Patient and public involvement

Patients and members of the public were involved at all stages from the development of the intervention during feasibility testing through to trial design and conduct. We received input from older adults on the design of the trial materials and in the oversight of the delivery of the study through PPI representation on trial oversight groups. The main trial results were disseminated to participants using materials suitable for a non-specialist audience and shared with the wider public, clinical and academic colleagues through events and publications.

### RESULTS

The 'Fidelity Assessment Framework' including evidence for fidelity is presented in table 1.

### Facilitators

All four facilitators remained the same throughout the trial at both sites. They were all female, younger than the youngest participant (65 years) and from a healthcare or social care professional background.

### Intervention attendance

Eleven Lifestyle Matters interventions were delivered, six in one city in Northern England and five in North Wales. Four groups were selected for the fidelity assessment as previously described. Overall attendance at the observed group sessions was 70% or above (see table 2). The average number of out-of-venue activities conducted as a part of the group aspect of the intervention across all groups was four.

### Training delivery

The 2-day training was attended by all four facilitators (two from each site) and the two supervisors (one from each site).

Excellent fidelity was demonstrated for trainer performance with a score of 100% across all four domains.

Good fidelity was identified through a criterion score of 75.4% across eight domains for trainees. Overall training fidelity was very good with a total agreed criterion score for 'trainer performance' and 'trainee performance' of 83.9%.

### Intervention delivery

#### Facilitator performance

Excellent or very good fidelity was achieved in four out of the eight recorded groups (see table 3). Only one observed group (Wave 1, Site A, Video 2) demonstrated unsatisfactory fidelity (52%). The other three groups all showed either an increase or maintenance in fidelity scores.

Excellent or very good fidelity was demonstrated in three of the six assessment domains. Only one domain 'Goals and Needs' was rated as being unsatisfactory. This required evidence of programme tailoring and how facilitators helped participants to select achievable activities with appropriate levels of challenge.

The decline in fidelity seen at Site A during Wave 1 (77% to 52%), could be explained by the facilitators arranging for two external speakers to deliver sessions on relaxation and physical health. During the meeting the facilitators largely left facilitation to the external speakers who used a didactic style, which did not invite discussion or tailoring to the needs of group members. It was deemed by both coders that the facilitators could have coached external speakers on how to deliver a session in accordance with the Lifestyle Matters ethos and could have interjected during the sessions. An outcome from this session was to provide facilitators with further guidance on when and how to engage external speakers. This was delivered as an email to all facilitators ensuring they received consistent information.

A difference was also noted between the first and last waves at Site B where the assessed fidelity dropped from 87.5% to 70.8% but was still within the boundary set for satisfactory fidelity. Based on findings from the qualitative interviews, reasons for the reduction in scores could be explained by difficulties in time management to undertake planning and delivery of multiple consecutive groups as well as the challenges presented by the dynamics within groups.

#### Group member performance

Excellent or very good fidelity was demonstrated for four out of the eight recorded groups (see table 4). Only one group (Wave 1, Site A) was rated as being unsatisfactory (41%) with this being related to the same issue described earlier for 'Facilitator performance'.

Excellent or very good fidelity was demonstrated for three out of eight identified domains with unsatisfactory evidence for only one, 'Confidence and hope'. However, it became evident that ability to score some criteria was affected by the nature of the activities participants engaged in.

#### Inter-rater reliability

Kappa scores were calculated using linear weights.[35]

For the assessment of training fidelity, the inter-rater reliability between coders was Kappa=0.72 (p<0.0.001), 95% CI 0.478 to 0.964.

Inter-rater reliability for overall fidelity to intervention delivery across sites was Kappa=0.63 (p<0.001), 95% CI 0.543 to 0.719.

Inter-rater reliability for facilitator domains across both sites was Kappa=0.55 (p<0.001), 95% CI 0.432 to 0.678.

| Table 3 | Fidelity scores for 'facilitator performance' | | | | | | | |
|---|---|---|---|---|---|---|---|---|
| | **Facilitator score/out of 48** | | | | | | | |
| | **Rater 1** | **Rater 2** | **Agreed score** | **%** | **Rater 1** | **Rater 2** | **Agreed score** | **%** |
| Wave 1 | (Site A) Group 1 | | | | (Site B) Group 1 | | | |
| Video 1 | 38 | 33 | 37/48 | 77.0% | 38 | 39 | 39/48 | 81.2% |
| Video 2 | 22 | 28 | 25/48 | 52.0% | 44 | 44 | 42/48 | 87.5% |
| Wave 3 | (Site A) Group 6 | | | | (Site B) Group 5 | | | |
| Video 1 | 44 | 44 | 44/48 | 91.6% | 30 | 34 | 30/48 | 62.5% |
| Video 2 | 46 | 45 | 44/48 | 91.6% | 33 | 35 | 34/48 | 70.8% |

**Table 4** Fidelity scores for 'group members'

| | Group member score/out of 39 | | | | | | | |
|---|---|---|---|---|---|---|---|---|
| | Rater 1 | Rater 2 | Agreed score | % | Rater 1 | Rater 2 | Agreed score | % |
| Wave 1 | (Site A) Group 1 | | | | (Site B) Group 1 | | | |
| Video 1 | 34 | 33 | 33/39 | 84.6% | 37 | 35 | 38/39 | 97.4% |
| Video 2 | 11 | 18 | 16/39 | 41.0% | 30 | 29 | 29/39 | 74.3% |
| Wave 3 | (Site A) Group 6 | | | | (Site B) Group 5 | | | |
| Video 1 | 33 | 31 | 33/39 | 84.6% | 23 | 28 | 24/39 | 61.5% |
| Video 2 | 36 | 36 | 36/39 | 92.3% | 26 | 26 | 26/39 | 66.6% |

For group members inter-rater reliability was Kappa=0.72 (p<0.001), 95% CI 0.598 to 0.844.

## Qualitative findings

The following themes relevant to intervention fidelity were identified from analysis of the qualitative interviews conducted with facilitators following cessation of delivery. Other qualitative findings including those derived from interviews with participants are reported elsewhere.[14 23]

### Theme 1: Experience as an enabler of intervention delivery

Analysis of facilitator interviews revealed that they all considered that they had become more confident, skilled and flexible over time. Also, that their commitment to the ethos of the programme was maintained:

> We've actually grown in confidence…it feels a very different place to where we were at the beginning… far more skilled… allowing for that flexibility and spontaneity (Facilitator D).

### Theme 2: Views of how skills acquisition for intervention delivery could have been enhanced

Further training opportunities such as booster sessions or shadowing experienced facilitators were suggested to help consolidate knowledge of the intervention and improve fidelity further.

> …it would have been useful just to have…a follow-up session just to see what kinds of things were coming up for people and have that, that kind of discussion (Facilitator B).

> Regular self-review, using a group observation checklist was proposed by facilitators as a tool to enable them to reflect on delivery style and decision-making processes. (Facilitators were aware of the checklist as part of the fidelity work but did not have sight of the checklist.)

> it may be more useful in future, if the facilitators could peer review themselves every 3 weeks against the checklist…scoring themselves and detailing an action against those that need to be achieved (Facilitator A).

### Theme 3: Need for more training to support greater understanding of group dynamics

One facilitator suggested that additional content in the training on how to avoid dependency relationships when facilitating using concrete examples case studies and written guidance would have been helpful.

> …what might breed relationships of dependency, what are the little things we (facilitators) don't realise we're doing, I think that might be something that could be added to the training (Facilitator A).

The same facilitator thought that fostering and enabling independence was also important for the intervention and more time dedicated in the training was needed.

> …to process and assimilate all that… (Facilitator A).

Another facilitator suggested that training should incorporate more on managing group dynamics and handling difficult people, and how to adapt sessions spontaneously when needed.

> …few trouble-shooting scenarios… (Facilitator D).

### Theme 4: Tailoring intervention components

One facilitator described modifying their approach to goal setting within the individual sessions where they felt participants had found the concept too formal or difficult to understand. Rather than talking about conscious goals they would integrate it subtly into the intervention.

> Interestingly you'd be approaching that same subject in a very subtle way over the weeks and weaving it into natural conversation, that was received a lot better… (Facilitator A).

### Theme 5: Anxiety as a consequence of researcher observations

Two facilitators reported levels of performance anxiety due to the session being recorded, which they felt may have hindered their delivery of the intervention.

> I think it is fair to say that being recorded has significantly affected the performance of the facilitators due to nerves of being recorded. As such I don't think the recordings reflect what has actually taken place within the groups over time. We tick many of the boxes on the checklist, but the recordings I feel won't pick a lot of this up (Facilitator D).

## DISCUSSION
### The framework

The fidelity assessment framework designed for this study provided a satisfactory structure for monitoring and evaluating adherence to the intervention over time. We were able to demonstrate good fidelity to training as well as delivery of the group component of the intervention but were unable to observe all intervention components including the individual sessions and the out of venue activities. It was significant, however, that participant adherence to the individual sessions was poor (32% of available sessions). As these sessions were deemed to be an essential component of the overall intervention, the reasons require further exploration. We were aware from the qualitative interview data that goal setting as part of the individual sessions proved problematic for facilitators to deliver. This was attributed to the significant tailoring needed to meet the needs of individual participants.

There were however several shortcomings to the framework approach[29] for this type of intervention. Assessment frameworks are designed to rate set criteria, which by definition should be the same for all observations. This is not necessarily true of complex interventions such as Lifestyle Matters that have tailoring as an underlying mechanism. To accommodate this, the underpinning ethos of the intervention along with the manual and training content provided the basis of the observation checklists.[39] This allowed for a range of observed behaviours to be considered in relation to set criteria instead of only one behaviour signifying presence.

### Fidelity assessment methodology and adaptation

We recognised the adaptive nature of the intervention and the resultant complexity for fidelity assessment at the time of delivering this study. Current debate is indeed questioning whether fidelity and adaptation can coexist.[40] Being able to determine the difference between intervention ineffectiveness or alternatively failure to deliver as intended can significantly impact the implementation outcome.[41] This intervention includes a number of interlinked adaptations, namely, selection of different components from a menu of possibilities; the context, the characteristics and needs of the participant and the group they are a member of; and the characteristics and abilities of facilitators. Whether it is possible to measure a range of adaptions for such tailored interventions, and how this would be achieved requires further consideration.

### Coding

Coding performance can raise concerns. Expectations of both the facilitators and those rating performances must be considered and moderated. All those involved should try to avoid preoccupation with fidelity assessment at the cost of intervention delivery.[42] Coders were not independent of the study, having attended the training as participant observers. Having the same knowledge and learning experience as those expected to deliver the intervention could enhance coder observations through greater understanding of observed behaviours and processes. Alternatively, using independent coders would reduce bias of unknown factors arising from personal knowledge of facilitators and increase the rigour of fidelity assessment. Overall we deduce that when the intervention is complex, ability to interpret what is occurring is required if accurate observations are to be recorded.[43] Using researchers with knowledge or experience of the intervention as coders may therefore be preferable.

### Training and competence

One of the main challenges to fidelity was the delivery of high-quality in-depth training for a complex psychosocial intervention in just 2 days to a range of staff. Chances of variation in intervention delivery were deemed to lie in facilitator background and experience which are related to fidelity.[6] Evidence from the qualitative interviews also indicated that coverage of several core principles of the intervention including managing individual and group dynamics could have been improved during training. In this trial it was not possible for facilitators to give feedback on the training they had received but in real world implementation we would recommend including this. To scaffold fidelity we employed senior members of staff with experience working with older people and behaviour change interventions to supervise facilitators in the delivery of the intervention.[44] Due to the novelty of the intervention, it was not possible to identify staff with existing experience of delivering it or anything similar. Therefore, a recommendation for future research would be to include evaluation of fidelity to supervision in more detail including competence and content. This has since been explored in another trial, Journeying through Dementia.[8]

The Lifestyle Matters manual and training are founded in 'occupational therapy' theory and practice and the facilitators were supervised by trained Occupational Therapists. However, in pursuance of evidence that the intervention might be delivered by a range of staff, the recruited facilitators came from a variety of professional backgrounds including social work and mental health advocacy as well occupational therapy. Further customisation of training may be required to meet the needs of a wider range of health and social care staff.

Some deterioration of skills, or drift, is common post training.[25] Although no additional training or booster sessions were provided during this study, facilitators received regular supervision and occasionally received emailed 'study updates'. These communications (approved by the Trial Management Group) clarified details on intervention delivery to help reduce potential facilitator drift. Any emergent ethical concerns would have been immediately acted on if identified during the fidelity assessment. Although fidelity to group intervention delivery was generally found to be good, further training opportunities such as booster sessions or shadowing experienced facilitators might be incorporated in future studies to test whether additional training

consolidates knowledge of the intervention and further improves fidelity. In addition the group observation checklist used to assess adherence and competence[13 45] to deliver the intervention could be used by facilitators as a tool to self-regulate and thereby enhance fidelity.[46]

### Facilitators and supervisors

It has been suggested that due to the key role of facilitators they should be cognisant of the impact of fidelity.[6] The importance and nature of the planned fidelity assessment was explained to the facilitators during the 2-day preparatory training before they took part in the assessment. This was beneficial in providing reassurance about the process including the observations.

Even though facilitators were from a range of backgrounds and had different experiences of working with older people in the community, this did not appear to negatively impact on intervention delivery.[34] The facilitator job description and person specification provided a priori level of credentials required to provide some consistency across facilitators. However, the previous point about responsiveness to needs for training and support by those running the study should be maintained. This also applies to supervisors who are not experienced in delivering the programme.

A 'Good fit' between providers and participants can be key to successful intervention delivery.[25] All facilitators in this trial were female and younger than the youngest participant but this did not appear to impact on delivery. The supervisors were generally older and more experienced in delivering occupational therapy-based interventions, which could be drawn on during supervision sessions.

### Receipt of intervention

Fidelity to the intervention increased between waves for Site A, but gradually decreased for Site B. This could have been due to the dynamics of the groups that were recorded; challenges with the methods used to assess fidelity or may have been indicative of facilitator drift over time. The only factor that could be identified to explain facilitator drift at site B was the time pressure they expressed in trying to prepare and deliver the group sessions as well as trying to organise and conduct individual sessions. This explanation is commensurate with the slight decline seen in 'Facilitator performance' and suggests that additional training may have been beneficial.

Fidelity was also influenced by receipt of the intervention. For example, we expected to observe group and individual goal-setting opportunities in intervention sessions. However, from the qualitative interviews it was evident that the majority of participants struggled to engage with the concept of 'goal setting' and facilitators were required to make significant changes to their own understanding and how they presented the concept to participants.[14] Low fidelity therefore may not necessarily be associated with poor fidelity to delivery but may highlight a problem with the original intervention requirements, the nature

of associated training and supervision, and subsequent criteria identified to evidence fidelity. Detailed analysis of feasibility study findings is therefore essential to establish clear baselines for behavioural interventions. Good design and piloting of tools is also important.

### Practicalities and ethical considerations

The assessment only involved the video recording of a small number of weekly meetings out of the total provided and therefore it is inevitable that some facilitator and participant behaviours that may have affected intervention delivery were not observed. The assessment process also did not include observations of facilitators arranging and conducting individual sessions or out-of-venue activities which would have enhanced our understanding of the underlying intervention mechanisms. As previously described there were ethical considerations around participant confidentiality when assessing fidelity in a community setting.[47] Consideration should therefore be given to how fidelity assessment can be undertaken within any given setting without interfering with the intervention.[48] How many, when and what type of sessions to record should be dependent on the complexity of the trial and availability of resources. For example, it may be prudent to monitor consecutive sessions to capture more information regarding participant receipt and enactment. Behaviour change may be more evident if an activity undertaken during one session can then be followed through to the next.

Psychosocial interventions are not designed to be passive; participants and facilitators bring their life experience, beliefs and expectations.[34 49] When determining intervention fidelity it is therefore important to consider how group dynamics and also environmental factors[50] impact not only on group composition but also group behaviour, especially when external factors to the intervention may be outside the control of those facilitating.[49]

The findings obtained from assessment of intervention fidelity within this RCT were important, given the neutral results obtained from analysis of outcome data.[10] The fidelity assessment enabled identification of the aspects of intervention delivery that were delivered as per protocol as well as those aspects that were compromised such as delivery of the individual sessions and specifically ability to facilitate goal setting with participants. These issues help inform our understanding and interpretation of the trial results as well as indicate the challenges that future implementation would involve.

### Conclusions

The outcomes from this fidelity assessment add to the growing literature on the use of fidelity assessment in trials and other studies of complex interventions. Psychosocial interventions that are flexible and offer a tailored approach in delivery may make fidelity assessment difficult, but not impossible. Paradoxically the importance of undertaking thorough fidelity assessment is enhanced for complex interventions, which have a number of

components. Despite the challenges, this study has demonstrated that by testing methods identified in a replicable framework, fidelity assessment can be carried out successfully.

Further research should explore the potential of different observational methods for monitoring fidelity in psychosocial community-based complex trials and the impact of these methods on intervention delivery.

**Acknowledgements** The authors thank the PPI contributors as well as Prof John Brazier, Dr Daniel Hind, Prof Stephen Walters, Dr Gill Windle, Prof Robert Woods (TMG) for advice on the trial protocol including the process evaluation and fidelity assessment; Dr Pip Logan, Dr Jennifer Wenborn, Dr Linda Sheppard, Dr Fiona Goudie of the Trial Steering Committee (TSC) and Dr Mona Kanaan, Prof Avril Drummond and Dr Claire Ballinger of the Data Monitoring and Ethics Committee (DMEC) for advice on and critical review of the trial protocol including the process evaluation and fidelity assessment.

**Contributors** GM and CC obtained the funding. GM, KS and CC developed the research design, fidelity assessment framework and data collection tools. KS managed the research and undertook data collection. KS and GM analysed and interpreted the data. KS drafted the manuscript. All authors critically reviewed and approved the manuscript.

**Funding** The trial was funded through the Medical Research Council (MRC) Lifelong Health and Wellbeing (LLHW) programme, grant number G1001406, ISRCTN is 67209155. The views and opinions expressed in this paper are those of the authors and do not necessarily reflect those of the MRC. Primary Care Research Network (PCRN) funding was accessed to support recruitment activity in GP surgeries in Sheffield and NISCHR provided support in North Wales. The funder has reviewed and approved the trial design including methods of data collection, analysis and interpretation of data.

**Competing interests** GM and CC are the original authors of the published Lifestyle Matters manualised programme. No other authors have any competing interests.

**Patient consent for publication** Not required.

**Ethics approval** The study was approved by the South Yorkshire Research Ethics Committee 12/YH/0101, the National Institute for Social Care and Health Research (NISCHR) Permissions Co-ordinating Unit in Wales, Betsi Cadwaladr University Health Board. Written informed consent was obtained from all participants.

**Provenance and peer review** Not commissioned; externally peer reviewed.

**Data availability statement** Data are available upon reasonable request. The data sets generated and analysed for this study will be available upon request from the authors.

**ORCID iD**
Kirsty Sprange http://orcid.org/0000-0001-6443-7242

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
