## [Reviewer comments · BMJ Open]

ARTICLE DETAILS

TITLE (PROVISIONAL)	Evaluation of intervention fidelity in a randomised controlled trial of a complex psychosocial intervention: Lifestyle Matters
AUTHORS	Sprange, Kirsty; Mountain, Gail; Craig, Claire

VERSION 1 – REVIEW

REVIEWER	Liz Steed Queen Mary University of London
REVIEW RETURNED	07-Sep-2020

GENERAL COMMENTS	I have read this paper with interest and concur with the authors that fidelity is an important issue when developing and evaluating complex interventions. Unfortunately this particular paper appears rather outdated with not only data collection conducted several years ago, but also the majority of references almost a decade out of date. There is little consideration of the recent progress that has been made in the area with publications such as that by Walton 2019 (https://doi.org/10.1111/bjhp.12394) and Toomey 2020 (https://doi.org/10.1080/21642850.2020.1738935) just a couple of the important recent contributions to the area. I also unfortunately find the ‘clarity of expression’ does not meet journal standards, equally I don’t find the tables particularly helpful in understanding what aspects of fidelity were maintained. Specific issues of concern include:- Abstract & Background The authors claim they develop a framework for assessing fidelity when actually they are using Bellg et al’s 2004 BCC consortium framework. It is actually a strength to use this framework to guide assessment, so I think the authors should refer to this far earlier in the manuscript and potentially in the abstract. There seem to be confusion between several concepts namely fidelity assessment, techniques to improve fidelity, maintenance of fidelity and causal mechanisms. The authors do not use these terms accurately at times leading a reader to be potentially misinformed. For example authors state that fidelity assessment can increase effect sizes and improve power – surely it is maintenance of fidelity which can do this not simply assessment. There is also discussion around causal mechanisms and their role in implementation. Understanding causal mechanisms is important and is often part of a process evaluation, which fidelity too falls into, but they are not the same and it is confusing to present them as if they are. Without clear understanding by the authors of the
--

	different ideas they are presenting there is a danger that a reader would actually become less informed after reading the paper. More background on the rationale for the intervention and content would be useful. For example it is not clear why a largely retired population would be offered an occupational intervention. Methods In the fidelity assessment there seems to be primarily a focus on adherence but little consideration of competence. This is a critical aspect of fidelity as discussed in Carroll 2007. A reflection of this and the limitations this gives to the current fidelity assessment should be explored. It is not apparent how the scoring system was devised e.g. why were present scores converted to '3'. Also on what basis were the percentage scores for fidelity decided? I am not sure on the validity of having a coder also be part of the training intervention, further justification of this and discussion on the potential for bias this brings in should be added in the discussion. How was it decided which sessions should be recorded and which groups – this should be explicit and consider whether any potential bias was brought in to play. Results I do not find the tables particularly informative it would be helpful in the text to have greater discussion on which aspects of fidelity were met to a greater or lesser extent One way the qualitative interviews could have informed fidelity assessment is if there was comment about actions taken which reflected receipt and use of training, however this does not appear to be present. Discussion is more around techniques to enhance fidelity, which whilst helpful, is slightly separate from an assessment of fidelity and it would be helpful for this to be noted. Discussion There is ongoing debate between the importance of fidelity versus adaptation for interventions that are to be implemented. There is no discussion of these important issues and how this paper fits within these, which would make this paper of greater interest.
--	---

REVIEWER	Prof Nusrat Husain University of Manchester, UK
REVIEW RETURNED	03-Dec-2020

GENERAL COMMENTS	Fidelity measure is most important to ensure robust scaling up of psychological and social interventions. This is often not reported and is a major limitation in implementation science. I have only a few comments and it would be helpful if the authors could provide some further details. I am finding it difficult to understand what was mentioned in “strengths and limitations” section - “it was not appropriate or even ethical to undertake observations of activities in the community”. It will be helpful if the authors could elaborate particularly around ethics.
--

	In background section authors made a strong rational for this study. However, some references need to be updated such as for “MRC recommendations for complex interventions”. On page number 7, line number 10; it would be useful to add reference for the manual that was used in the trial. Authors should describe facilitators load (how many participants/groups per facilitator at a given time) as it may effect quality of intervention delivery and hence fidelity. Video recording of training was done with consent of facilitators and trainers? How many facilitators in total were involved? Were all facilitators’ performance was rated? Supervision is critical across all areas but is particularly important in task shifting and task sharing. If the authors could report what was the frequency of supervision sessions and format of supervision? Content checklist should be included in appendices. In qualitative section authors need to explain the use of topic guide, who conducted interviews, why only 10% participants were selected for interviews? It would be interesting to know whether there were any differences in fidelity between group vs individual sessions. Typos page 7 line 17 “together to on future” “findings It was estimated” “For the purposes of the fidelity assessment, data from the interviews to identify cross cutting themes and explanatory factors for fidelity outcomes” Overall in my opinion it is a well written manuscript and will be a useful addition to the literature.
--	--

VERSION 1 – AUTHOR RESPONSE

Reviewer 1	
There is little consideration of the recent progress that has been made in the area with publications such as that by Walton 2019 (https://doi.org/10.1111/bjhp.12394) and Toomey 2020 (https://doi.org/10.1080/21642850.2020.1738935) just a couple of the  important recent	We have revised the manuscript, particularly the discussion to reflect current literature and science in this area.
Abstract & Background The authors claim they develop a framework for assessing fidelity when	We have revised the references accordingly on page 7 and added a

actually they are using Bellg et al's 2004 BCC consortium framework. It is actually a strength to use this framework to guide assessment, so I think the authors should refer to this far earlier in the manuscript and potentially in the abstract.	ref as a footnote to Table 1. We have also referred to the framework in the abstract.
There seem to be confusion between several concepts namely fidelity assessment, techniques to improve fidelity, maintenance of fidelity and causal mechanisms. The authors do not use these terms accurately at times leading a reader to be potentially misinformed. For example authors state that fidelity assessment can increase effect sizes and improve power – surely it is maintenance of fidelity which can do this not simply assessment. There is also discussion around causal mechanisms and their role in implementation. Understanding causal mechanisms is important and is often part of a process evaluation, which fidelity too falls into, but they are not the same and it is confusing to present them as if they are. Without clear understanding by the authors of the different ideas they are presenting there is a danger that a reader would actually become less informed after reading the paper.	We have revised the wording on page 3-4 of the 'Background' section to clarify terminology used.
More background on the rationale for the intervention and content would be useful. For example it is not clear why a largely retired population would be offered an occupational intervention.	We have included more detail on the rationale for an occupational intervention on pages 4-5 in the Background section.

In the fidelity assessment there seems to be primarily a focus on adherence but little consideration of competence. This is a critical aspect of fidelity as discussed in Carroll 2007. A reflection of this and the limitations this gives to the current fidelity assessment should be explored.	We have expanded the discussion, see section now titled 'Training and competence'.
It is not apparent how the scoring system was devised e.g. why were present scores converted to '3'. Also on what basis were the percentage scores for fidelity decided?	More detail has been added to page 10 describing the scoring system based on Borelli 2011.
I am not sure on the validity of having a coder also be part of the training intervention, further justification of this and discussion on the potential for bias this brings in should be added in the discussion	In the discussion on page 19 under the header 'Coding' we believe we already discuss the issue of bias arising from whether to use independent coders or those with

1

	knowledge of the intervention to inform observations. We have however extended this discussion point.
How was it decided which sessions should be recorded and which groups – this should be explicit and consider whether any potential bias was brought in to play.	More detail on selection of groups/sessions to be recorded/observed is included on page 11 under the heading 'Assessment of the fidelity of intervention delivery'.

I do not find the tables particularly informative it would be helpful in the text to have greater discussion on which aspects of fidelity were met to a greater or lesser extent	We have suggested removing table 3, since the text below the table describes the key information. We have also included a brief summary of the overall findings at the start of the discussion on page 18.
One way the qualitative interviews could have informed fidelity assessment is if there was comment about actions taken which reflected receipt and use of training, however this does not appear to be present. Discussion is more around techniques to enhance fidelity, which whilst helpful, is slightly separate from an assessment of fidelity and it would be helpful for this to be noted.	We have included more detail from the qualitative analysis in the Results (page 17) and Discussion sections regarding facilitator reflection on the training and its impact on their ability to deliver the intervention.
Discussion: There is ongoing debate between the importance of fidelity versus adaptation for interventions that are to be implemented. There is no discussion of these important issues and how this paper fits within these, which would make this paper of greater interest.	We have expanded the discussion section to include details of this debate. See section titled 'Fidelity assessment methodology and adaptation' on page 17.
Reviewer 2	
I am finding it difficult to understand what was mentioned in “strengths and limitations” section - “it was not appropriate or even ethical to undertake observations of activities in the community”. It will be helpful if the authors could elaborate particularly around ethics.	More detail on the ethical considerations has been added to the section on ‘practicalities’ on page 21.

In background section authors made a strong rational for this study. However, some references need to be updated such as for “MRC recommendations for complex interventions”.	The background section has been revised and references updated.
On page number 7, line number 10; it would be useful to add reference for the manual that was used in the trial.	We have included a reference to the published manual on page 7 under the ‘Training’ header.
Authors should describe facilitators load (how many participants/groups per facilitator at a given time) as it may effect quality of intervention delivery and hence fidelity.	A description of the facilitator workload has been added to page 7 under the ‘Facilitators and facilitation’ header.
Video recording of training was done with consent of facilitators and trainers? How many facilitators in total were involved? Were all facilitators’ performance was rated?	More detail has been added to page 13 under the ‘Training delivery’ header.
Supervision is critical across all areas but is particularly important in task shifting and task sharing. If the authors could report what was the frequency of supervision sessions and format of supervision?	More detail has been added to page 7 under the ‘Facilitator supervision’ header.
Content checklist should be included in appendices.	There is now only one additional file for the submission which contains two documents, the two checklists, one for training and one for the

2

	group sessions. We have added a contents list to this document for clarity.
--	--

In qualitative section authors need to explain the use of topic guide, who conducted interviews, why only 10% participants were selected for interviews?	More detail has been added to page 13 under the 'Semi-structured interviews' heading.
It would be interesting to know whether there were any differences in fidelity between group vs individual sessions.	We are unable to provide this data as individual sessions were not observed as part of the fidelity assessment. We have clarified this point on page 12.
Typos page 7 line 17 "together to on future" "findings It was estimated" "For the purposes of the fidelity assessment, data from the interviews to identify cross cutting themes and explanatory factors for fidelity outcomes"	All errors have been corrected.
Editor	
Table/s should be embedded Kindly embed your table (should be editable and in table tools format). Tables should be placed in the main text where the table is first cited.	All tables are now embedded in the manuscript.
Please re-upload your supplementary files in PDF format.	We have merged all supplementary files into one document with a contents list and re-uploaded.
Figure/s should not be embedded Please remove all your figures in your main document and upload each of them separately under file designation 'Image' (except tables and please ensure that figures are in better quality or not pixelated when zoomed in).	There are no figures in the manuscript.

VERSION 2 – REVIEW

REVIEWER	Steed, Liz Barts and The London School of Medicine and Dentistry, Centre for Primary Care and Public Health
REVIEW RETURNED	26-Feb-2021

GENERAL COMMENTS	This paper has been significantly improved since it's first submission and is now well written and has up to date references. I feel it could be a helpful addition to the literature. I do still have some concerns however, particularly in relation to what the trial outcomes were and how this may or not relate to the fidelity assessment. I also have concerns about what the qualitative aspect is purported to do. These and other comments are outlined below. Introduction 1. It is stated that lifestyle matters is an 'occupational' intervention. It would be useful to provide a definition of what occupational means in this context – e.g. is it specifically about paid for work or more diverse activities. Presumably the later given this is an intervention for over 65s in which case it is even more important to describe what is meant by occupational activities. 2. I have concerns about the description of what the qualitative work aimed to do i.e. you say 'exploration of the mechanisms that mediated or moderated lifestyle change in study participants; thereby identifying causal mechanisms associated with implementation' (pg 4, ln 31-40). I think suggesting the qualitative work explores causal and mediator and moderator roles is rather overstating the role of this work. In addition, mediator and moderator are common statistical terms for describing relationships between intervention outcomes and processes. In my opinion it would be preferable to present the qualitative work as simply a complementary (but still important and useful) method to expand understanding of intervention delivery by gaining the facilitators perspective on what could have improved the intervention and things they found helpful or difficult in the delivery. I do not feel in any way the small amount of data presented represents analysis of causal mechanisms. 3. On page 5 ln 24/25 you mention the discussion of fidelity will be related to intervention implementation. It is not mentioned whether the intervention was shown to be effective and how fidelity relates to this. I think this would be of interest to the readers. Indeed on looking up publication 11 it appears there was no clinical or cost effectiveness demonstrated so it would not be appropriate to go ahead with implementation. Discussing fidelity in the context of this non-significant finding is still important as it allows you to evaluate one possible reason for this lack of effect. I think such discussion could be a helpful addition to the manuscript. Methods 4. Pg 6 – says facilitators were recruited with required skills but the reference for this is then Borelli fidelity framework. Could the authors be more explicit in the exact skills they required and how they determined these were necessary.
--

	5. Supervision is a strength – I wondered if there was any information on the outcome of this. 6. At what point was the fidelity framework developed, could this be made a little more explicit. Results 7. Pg 6 Ln 17 States – ‘A crucial element is going into the community to try out new or neglected skills with support from each other and from the facilitators; at least four sessions should involve such activities.’ However only 4% of individuals achieved this which says to me the intervention was not delivered/received as intended. I don’t feel this comes across in the conclusions. Rather the manuscript says that the intervention was delivered as intended in the abstract. 8. Similarly I think it is of note that one aspect of fidelity which you mention that was not delivered as intended is the goal setting. This is an important element of behaviour change so if this was not delivered it may give an important clue as to why the intervention did not work. I think this and the above in point 7 should be reflected on more in the discussion as it has important implications for how future interventions are designed. 9. The approach to assessing training delivery by being a participant is an interesting one. It allows the researcher to get a feel for some of the more subtle elements of delivery such as group dynamics etc, however it does suffer from some disadvantages in that by being a participant one can take a less objective and panoramic perspective. I think further reflection on this in the discussion would be useful. 10. It is interesting that the inter-rater coding was at the lowest end of good. Did the authors take any actions to improve inter-rater reliability throughout the coding process. 11. I was unclear whether there was feedback after wave 1 to support the delivery of the intervention i.e. was the fidelity assessment only descriptive or was there a formative element as well? 12. The qualitative findings are very brief with no themes presented. In the methods you mention framework analysis. If there was a change in analysis plan this should be mentioned and reasons discussed. 13. In the methods you mention interviewing participants yet I do not see any presentation of the results. I think this needs to be aligned with either methods saying this is reported elsewhere or presentation of findings in this paper. Discussion 14. One major limitation of the fidelity assessment is that it is focussed on adherence and does not reflect quality or competence. Whilst this may have been beyond the resources of the fidelity assessment and therefore not possible I do feel it is important to reflect on this limitation in the discussion.
--	---

	15. As noted above I think more discussion of which bits were delivered with fidelity and which bits weren't and how this potentially relates to the outcome of the trial would be useful.
--	--

REVIEWER	Husain, Nusrat The University of Manchester, School of Medicine
REVIEW RETURNED	09-Mar-2021

GENERAL COMMENTS	I want to thank authors for addressing comments in the revised manuscript. I have few observations on revised manuscript that I want to highlight; To avoid confusion it would be better to take out lines about the causal mechanisms as this manuscript is focusing only on fidelity and there is no information on causal mechanisms in methods, results and discussion. In “practicalities” authors have now explained that “there were ethical considerations around participant confidentiality when assessing fidelity in a community setting”. Can we add a reference here describing why this is an ethical concern? In background section there are still a number of references that are old such as on page 3 paragraph 2 in background section authors cited reference from 2004 and 2007 to explain fidelity assessment (Toomey, E., Hardeman, W., Hankonen, N., Byrne, M., McSharry, J., Matvienko-Sikar, K. and Lorencatto, F., 2020. Focusing on fidelity: narrative review and recommendations for improving intervention fidelity within trials of health behaviour change interventions. Health psychology and behavioral medicine, 8(1), pp.132-151.) On page 13 under the heading “semi-structured interviews” you have added a rationale for selection of 10% of the sample and you talked about data saturation. It would be good to add a reference here (Braun, V. and Clarke, V., 2019. To saturate or not to saturate? Questioning data saturation as a useful concept for thematic analysis and sample-size rationales. Qualitative Research in Sport, Exercise and Health, pp.1-16.).
---

VERSION 2 – AUTHOR RESPONSE

Reviewer 1	
1. It is stated that lifestyle matters is an ‘occupational’ intervention. It would be useful to provide a definition of what occupational means in this context – e.g. is it specifically about paid for work or more diverse activities. Presumably the later given this is an intervention for over 65s in which case it is even more important to describe what is meant by occupational	We have provided further clarification in the abstract and on page 4.

activities.	
2. I have concerns about the description of what the qualitative work aimed to do i.e. you say ‘exploration of the mechanisms that mediated or moderated lifestyle change in study participants; thereby identifying causal mechanisms associated with implementation’ (pg 4, ln 31-40). I think suggesting the qualitative work explores causal and mediator and moderator roles is rather overstating the role of this work. In addition, mediator and moderator are common statistical terms for describing relationships between intervention outcomes and processes. In my opinion it would be preferable to present the qualitative work as simply a complementary (but still important and useful) method to expand understanding of intervention delivery by gaining the facilitators perspective on what could have improved the intervention and things they found helpful or difficult in the delivery. I do not feel in any way the small amount of data presented represents analysis of causal mechanisms.	We have revised the manuscript to focus how the fidelity assessment was supported by qualitative evidence. We have referenced the papers that fully report the embedded qualitative study.
3. On page 5 ln 24/25 you mention the discussion of fidelity will be related to intervention implementation. It is not mentioned	We have added more detail to the discussion.

whether the intervention was shown to be effective and how fidelity relates to this. I think this would be of interest to the readers. Indeed on looking up publication 11 it appears there was no clinical or cost effectiveness demonstrated so it would not be appropriate to go ahead with implementation. Discussing fidelity in the context of this non-significant finding is still important as it allows you to evaluate one possible reason for this lack of effect. I think such discussion could be a helpful addition to the manuscript.	
4. Pg 6 – says facilitators were recruited with required skills but the reference for this is then Borelli fidelity framework. Could the authors be more explicit in the exact skills they required and how they determined these were necessary.	The Borelli paper is referenced as it discusses the importance of preserving fidelity through the right “fit between the provider and the

2

	population”, so recruiting the right people is essential. We have added more detail on the recruitment process and skills required as requested.
5. Supervision is a strength – I wondered if there was any information on the outcome of this.	We have included further information on supervision in the results section and discussion.
6. At what point was the fidelity framework developed, could this	We have clarified when the

be made a little more explicit.	framework was developed on page 8.
7. Pg 6 Ln 17 States – ‘A crucial element is going into the community to try out new or neglected skills with support from each other and from the facilitators; at least four sessions should involve such activities.’ However only 4% of individuals achieved this which says to me the intervention was not delivered/received as intended. I don’t feel this comes across in the conclusions. Rather the manuscript says that the intervention was delivered as intended in the abstract.	We are unsure where the 4% figure has been identified. On page 14 under the header ‘Intervention attendance’, we state that on average groups had attended four out of venue sessions which would meet intervention delivery as intended (described on page 6).
8. Similarly I think it is of note that one aspect of fidelity which you mention that was not delivered as intended is the goal setting. This is an important element of behaviour change so if this was not delivered it may give an important clue as to why the intervention did not work. I think this and the above in point 7 should be reflected on more in the discussion as it has important implications for how future interventions are designed.	We have updated the discussion section.
9. The approach to assessing training delivery by being a participant is an interesting one. It allows the researcher to get a feel for some of the more subtle elements of delivery such as	We have expanded on the observation method selection under the heading ‘Assessment of the

group dynamics etc, however it does suffer from some disadvantages in that by being a participant one can take a less objective and panoramic perspective. I think further reflection on this in the discussion would be useful.	fidelity of training delivery’.
10. It is interesting that the inter-rater coding was at the lowest end of good. Did the authors take any actions to improve inter- rater reliability throughout the coding process.	The coders undertook a review of the coding frameworks during the piloting process as described on page 11.
11. I was unclear whether there was feedback after wave 1 to support the delivery of the intervention i.e. was the fidelity assessment only descriptive or was there a formative element as well?	We believe we already discuss this point in the discussion on page 21.
12. The qualitative findings are very brief with no themes presented. In the methods you mention framework analysis. If there was a change in analysis plan this should be mentioned and reasons discussed.	We have now included the theme headers and referenced the qualitative findings which are published elsewhere for the trial. See page 16 for additional text.
13. In the methods you mention interviewing participants yet I do not see any presentation of the results. I think this needs to be aligned with either methods saying this is reported elsewhere or presentation of findings in this paper.	As above for point 11 we have now referenced the qualitative findings which are published elsewhere for

	the trial. See page 16 for additional
--	--

	text.
14. One major limitation of the fidelity assessment is that it is focussed on adherence and does not reflect quality or competence. Whilst this may have been beyond the resources of the fidelity assessment and therefore not possible I do feel it is important to reflect on this limitation in the discussion.	We believe we have already covered this subject under the header 'Training and competence' in the discussion.
15. As noted above I think more discussion of which bits were delivered with fidelity and which bits weren't and how this potentially relates to the outcome of the trial would be useful.	We have addressed this point through edits made in response to the above points.
Reviewer 2	
To avoid confusion it would be better to take out lines about the causal mechanisms as this manuscript is focusing only on fidelity and there is no information on causal mechanisms in methods, results and discussion.	We have addressed this point through edits made in response to reviewer 1.
In "practicalities" authors have now explained that "there were ethical considerations around participant confidentiality when assessing fidelity in a community setting". Can we add a reference here describing why this is an ethical concern?	We have included a reference as suggested.
In background section there are still a number of references that are old such as on page 3 paragraph 2 in background section authors cited reference from 2004 and 2007 to explain fidelity assessment	We have updated the reference as suggested.

On page 13 under the heading “semi-structured interviews” you

have added a rationale for selection of 10% of the sample and you

talked about data saturation. It would be good to add a reference

here

We have updated the reference as suggested.